# Cobalt catalyst with exclusive metal-centered chirality for asymmetric photocatalysis

Su-Yang Yao[1,2], Marco Villa [3], Yuan Zheng [1], Antonio Fiorentino [3], Barbara Ventura[4], Sergei I. Ivlev [1], Paola Ceroni [3] ✉ & Eric Meggers [1] ✉

For decades, progress in chiral transition metal catalysis has been closely linked to the design of tailor-made chiral ligands. Recently, an alternative to this conventional paradigm has emerged in which the overall chirality of the catalysts arises solely from a stereogenic metal center. However, the development of such chiral-at-metal catalysts based on earth-abundant metals is still a formidable challenge. Here, we report a reactive chiral-at-cobalt catalyst comprised entirely of achiral ligands, more than a century after Alfred Werner first introduced chiral cobalt complexes with exclusive metal-centered chirality. The cobalt center uniquely serves multiple functions: it is the sole stereocenter, redox center, catalytic site, and a chromophore. While the cobalt(III) complex is inert and bench-stable under ambient conditions, it can be photoactivated through an unexpected counterion-assisted mechanism, reducing inert cobalt(III) to catalytically active cobalt(II). This chiral-at-cobalt complex enables the visible-light-activated enantioselective conversion of isoxazoles into chiral 2H-azirines, achieving high enantiomeric excess of up to 97%.

The development of homogeneous catalysts based on earth-abundant metals, rather than noble metals, represents a significant advancement toward achieving more sustainable chemistry[1–4]. Among these, cobalt stands out as an attractive candidate due to its versatile organometallic chemistry, ability to access multiple oxidation states[5–8], and emerging prominence in photocatalysis[9–12]. Recent studies demonstrate the effectiveness of cobalt complexes as photocatalysts, revealing their ability to promote distinctive reactivity through unique mechanistic pathways under light irradiation. This photoactive behavior positions cobalt as a promising alternative to noble metals in light-driven transformations, opening innovative avenues for catalysis design[13–24]. In the realm of asymmetric catalysis, chiral cobalt catalysts have garnered considerable attention, with prominent examples including chiral cobalt porphyrins, chiral cyclopentadienyl complexes, chiral pincer complexes, and cobalt complexes featuring chiral phosphines or chiral N-heterocyclic carbene (NHC) ligands[25–27]. Despite this

progress, there remains a striking absence of reactive chiral cobalt catalysts constructed exclusively from achiral ligands[28]. This absence is particularly surprising given Alfred Werner's groundbreaking work over a century ago, which first introduced chiral cobalt complexes exhibiting exclusively metal-centered chirality[29]. The challenge in designing chiral-at-cobalt complexes lies in the intrinsic properties of cobalt's oxidation states. Octahedral cobalt(III) complexes, while geometrically well-suited for generating metal-centered chirality, are typically highly inert and thus unreactive. In contrast, cobalt(II) complexes, though reactive, suffer from lability, which can lead to the rapid loss of stereochemical information. This delicate interplay between reactivity and stability has likely contributed to the lack of viable chiral-at-cobalt catalysts in the literature, rendering the realization of such systems particularly challenging.

Here, we report a reactive chiral cobalt catalyst composed entirely of achiral ligands (Fig. 1). The cobalt-centered chirality arises from the

[1]Fachbereich Chemie, Philipps-Universität Marburg, Marburg, Germany. [2]School of Chemistry and Materials Science, Guangdong University of Education, Guangzhou, PR China. [3]Department of Chemistry "Giacomo Ciamician", University of Bologna, Bologna, Italy. [4]Institute for Organic Synthesis and Photo-reactivity (ISOF), National Research Council (CNR), Bologna, Italy. ✉e-mail: paola.ceroni@unibo.it; meggers@chemie.uni-marburg.de

**Fig. 1 | Chiral-at-cobalt asymmetric photocatalysis by light-induced counter-anion assistance.** BArF tetrakis(3,5-bis(trifluoromethyl)phenyl)borate, Mes mesityl, ee enantiomeric excess.

octahedral coordination of an unsymmetric meridional tridentate ligand, combined with one bidentate ligand and one monodentate ligand. The resulting chiral-at-cobalt(III) (pre)catalyst is remarkably stable, remaining inert and bench-stable in the dark without decomposition. Upon exposure to visible light, however, an unanticipated activation mechanism is triggered. This process involves MeCN dissociation, followed by a reduction to the reactive cobalt(II) oxidation state, mediated by the tetrakis(3,5-bis(trifluoromethyl)phenyl)borate (BArF) counteranion. The resulting in situ generated pentacoordinate chiral-at-cobalt(II) complex exhibits high catalytic activity and enantioselectivity in the conversion of isoxazoles to their chiral 2H-azirines.

## Results

### Design and synthesis of complexes

Over the past decade, our group[30–32] and others[33–41] have demonstrated the broad utility of chiral-at-metal catalysts in asymmetric catalysis[42,43]. Previous designs primarily relied on generating a stereogenic metal center through a propeller-type arrangement of two inert achiral bidentate ligands and two monodentate ligands in an octahedral coordination sphere $(2 + 2 + 1 + 1$ design)[43], while a few alternatives have been reported[33,36,41]. As an unrealized coordination mode for chiral-at-metal catalysts, we envisioned generating a stereogenic metal center using one inert meridional tridentate ligand, one inert bidentate ligand, and one labile monodentate ligand in an octahedral coordination sphere $(3 + 2 + 1$ design) (Fig. 2a). This arrangement creates a stereogenic metal center if the tridentate pincer ligand is unsymmetric. The higher denticity of the tridentate ligand is expected to enhance the inertness of the coordination sphere, a crucial feature for the development of chiral-at-metal catalysts based on 3d-metals. To date, such $(3 + 2 + 1)$ systems have only been realized with at least one chiral ligand[44,45]. Here, we present a chiral-at-metal catalyst based exclusively on achiral ligands within a $(3 + 2 + 1)$ coordination geometry.

The realization of such a $(3 + 2 + 1)$ coordination sphere is shown in Fig. 2b. Cobalt in the oxidation state +III is coordinated by one pincer-based pyridyl-2,6-bis-N-heterocyclic carbene (NHC) in addition to one N-(2-pyridyl)-substituted NHC and one MeCN ligand in an octahedral fashion. The three NHC ligands generate a strong ligand field and are supposed to contribute to a high configurational stability of the complexes. Since the two NHC moieties of the pincer ligand bear different substituents (methyl vs. aryl group), the cobalt center is stereogenic and the overall cobalt complexes are therefore chiral although all ligands are achiral. The tricationic complexes are complemented by one hexafluorophosphate and two tetrakis(3,5-bis(trifluoromethyl)phenyl)borate (BArF) counterions. The bulky and hydrophobic BArF counterions have the function to increase solubility of the salts in organic solvents.

The racemic complexes *rac*-**CoBr1-3** were synthesized via standard coordination chemistry, starting from CoBr$_2$ and sequentially coordinating the tridentate and bidentate ligands (see Supplementary Information, pages 5-16). Conversion to enantiomerically pure complexes was accomplished using a monodentate chiral auxiliary ligand approach (Fig. 3)[46–50]. Specifically, *rac*-**CoBr1-3** were reacted with (S)-methoxyphenylacetic acid ((S)-**Aux**) in the presence of K$_2$CO$_3$, followed by anion exchange with NaBArF to yield the two diastereomers Λ-(S)-**CoAux1-3** (28–35% yield) and Δ-(S)-**CoAux1-3** (30–36% yield), which were separated by silica gel chromatography. $^1$H NMR analysis confirmed the high diastereomeric purity of each diastereomer (≥98:2 dr). Treating the isolated diastereomers with NH$_4$PF$_6$ in MeCN at 70 °C yielded the corresponding enantiomers Λ-**CoCat1-3** (81–86%) and Δ-**CoCat1-3** (82–86%). The enantiomeric purity of these chiral-at-cobalt complexes was determined to be ≥97:3 er via $^1$H NMR analysis using Δ-TRISPHAT[51] as a chiral NMR shift reagent (see Supplementary Information, pages 33–37).

The crystal structure of a cobalt complex derivative (Δ-**CoCat4**) is shown in Fig. 4. Δ-**CoCat4** is a variant of **CoCat3**, where the CF$_3$ group on the PyNHC ligand is replaced by a 2,6-Me$_2$Ph group, and the complex is paired with three hexafluorophosphate counterions. Although **CoCat4** (with or without BArF counterions) proved catalytically inactive, presumably due to high steric hindrance, it was the only complex in our hands that yielded suitable crystals for single-crystal X-ray diffraction analysis. The structure confirms the $(3 + 2 + 1)$ coordination sphere, consisting of one tridentate, one bidentate, and one monodentate ligand arranged octahedrally and containing a stereogenic cobalt center. Notably, two interligand π-π stacking interactions likely contribute to the overall stability: the mesityl group of the bidentate pyridyl-NHC ligand stacks with the central pyridyl moiety of the pincer ligand, while the pyridyl moiety of the pyridyl-NHC ligand stacks against the central phenyl group of the N-aryl substituent on the pincer ligand. The crystal structure also highlights an important design rationale for selecting a pyridyl NHC ligand as the bidentate fragment: steric factors require that the NHC coordinates *trans* to the monodentate ligand (the reactive site), enabling the mesityl group to stack favorably with the π-system of the tridentate ligand and thus governing the diastereoselectivity. In the alternative theoretical diastereomer, the mesityl group would clash with the substituents of the tridentate ligand, making this arrangement significantly less favorable. In this context, it is noteworthy that attempts to coordinate 2,2′-bipyridine or 1,10-phenanthroline as bidentate ligands in place of the pyridyl-NHC were unsuccessful. In contrast, coordination of 2,2′-biquinoline yielded an achiral pentacoordinate complex (see Supplementary Information, pages 66-68), further underscoring the significance of the pyridyl-NHC moiety.

### Catalytic application

As expected for hexacoordinated cobalt(III) complexes, **CoCat1-3** are highly robust and remain stable when heated in MeCN at 70 °C for 24 h without any signs of decomposition. While this stability seems unfavorable for catalytic applications, we made the surprising discovery that this class of chiral-at-cobalt complexes catalyzes the light-induced conversion of isoxazole **1** to 2H-azirine **2** (Table 1)[44,45,52–55]. For example,

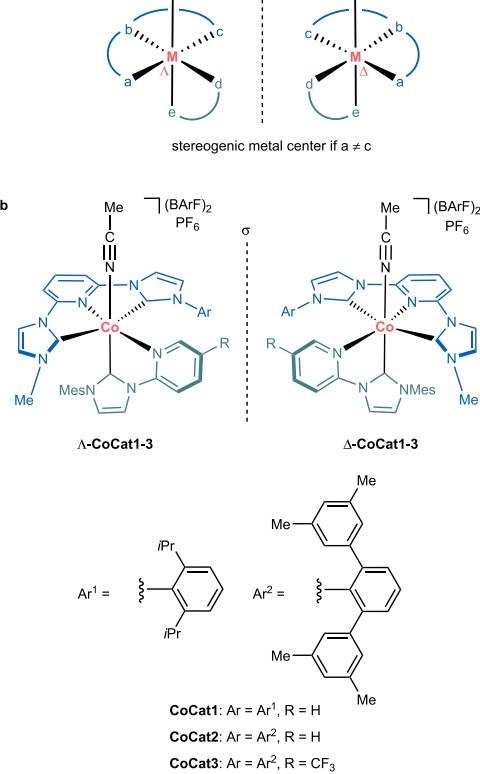

**Fig. 2 | New design strategy for chiral-at-metal catalysts using an unsymmetrical pincer ligand. a** Visualization of the design strategy. **b** Implementation of the strategy for the design of chiral-at-cobalt catalysts. BArF tetrakis(3,5-bis(trifluoromethyl)phenyl)borate, Mes mesityl.

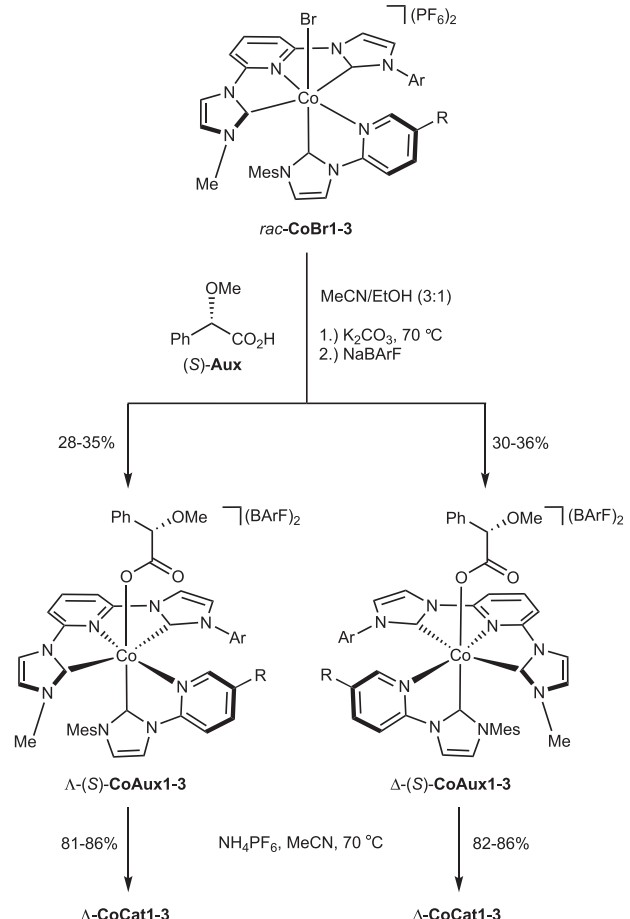

**Fig. 3 | Auxiliary-mediated synthesis of non-racemic chiral-at-cobalt complexes.** BArF tetrakis(3,5-bis(trifluoromethyl)phenyl)borate, Mes mesityl.

using Δ-**CoCat1** (3 mol%) in CH₂Cl₂ at room temperature under irradiating with blue LEDs (24 W) for 24 h provided azirine **2** in 90% NMR yield (entry 1). In contrast, only trace amounts of product were observed in the absence of light (entry 2). Unfortunately, the reaction yielded racemic product for this initial photoreaction. Reducing the temperature to −30 °C slightly improved the enantioselectivity to 2% ee, albeit with a reduced yield of 9% (entry 3). To improve the enantioselectivity, structural modifications of the cobalt complex were investigated. Modifying the tridentate ligand by replacing the 2,6-diisopropylphenyl group with a bulkier 2,6-bis(3,5-dimethylphenyl)phenyl moiety (Δ-**CoCat2**) improved the enantioselectivity to 42% ee, though with a yield of only 23% (entry 4). Introducing a CF₃ group into the bidentate ligand (Δ-**CoCat3**) further enhanced the yield to 93% and the enantioselectivity to 64% ee (entry 5). Further lowering the temperature to −45 °C improved the enantioselectivity to 82% ee (entry 6), and at −50 °C, it increased slightly to 83% ee but with a decreased yield of just 58% (entry 7). Interestingly, the addition of NaBArF (0.1 equiv) increased the yield to 96% (entry 8). Optimal conditions were achieved using Δ-**CoCat3** at a higher catalyst loading (6 mol%) at −60 °C with 0.1 equiv of NaBArF, affording the azirine in 78% yield with a satisfactory ee of 91% (entry 9). Finally, employing the mirror-image catalyst Λ-**CoCat3** (containing three BArF counterions) (entry 10) provided the enantiomeric azirine product with the same level of enantioselectivity (see comparison of entries 8 and 10). A brief substrate scope is shown in Fig. 5. and demonstrates that this catalytic asymmetric rearrangement is applicable to substrates with modified electronic and steric properties (**2b–j**, 48–81% yield, 79–97% ee).

## Mechanistic investigation
To elucidate the underlying photochemical mechanism, we investigated the photophysical and photochemical properties of these cobalt complexes. The complex *rac*-**CoCat3** exhibits intense ligand-to-metal charge transfer (LMCT) transitions in the UV range, characteristic of low-spin Co(III) complexes coordinated to aromatic ligands[56,57] and metal centered (MC) transitions in the visible spectral region (Fig. 6a): the spin-allowed transition to the $^1$MC state is peaked at ca. 415 nm ($\varepsilon_{415}$ $_{nm}$ = 350 M$^{-1}$ cm$^{-1}$), while the weaker band tailing up to 700 nm is attributed to the spin-forbidden transition to the $^3$MC state ($\varepsilon_{580}$ $_{nm}$ = 10 M$^{-1}$ cm$^{-1}$)[19,56]. The complex is not emissive at room temperature and at 77 K, as expected from the fast non-radiative decay typically exhibited by MC excited states of Co(III) complexes. The excited-state dynamics of the complex was studied by femtosecond transient absorption spectroscopy (Fig. 6b). Upon photoexcitation at 350 nm, a broad spectrum with a main absorption band in the 700-800 nm range is observed. This signal, which can be attributed to the lowest triplet state of the Co(III) complex[57], decays to zero with a time constant of 14 ps (Fig. 6b, inset). The very short lifetime of the lowest excited state of *rac*-**CoCat3** rules out the possibility of efficient bimolecular quenching processes. Consequently, we focused on ligand photodissociation, a well-established phenomenon in coordination chemistry[58–64].

Specifically, we exposed *rac*-**CoCat1** to an excess of CD₃CN in CD₂Cl₂ and monitored the exchange of MeCN with CD₃CN using $^1$H NMR spectroscopy (Fig. 7). As anticipated for inert Co(III) complexes, the ligand exchange proceeded very sluggish under ambient conditions, with a rate constant of $k = 1.67 \times 10^{-5}$ min$^{-1}$, corresponding to only 15% ligand exchange after 7 days at room temperature. Remarkably, upon irradiation with blue LEDs (24 W), the rate of ligand exchange increased significantly, achieving 31% exchange within just 2 h (under the experimental conditions used, an apparent rate

constant of $k = 3.23 \times 10^{-3}$ min$^{-1}$ was measured, which is an enhancement of approximately 200-fold). These results demonstrate that light exposure effectively acts as a switch to trigger ligand lability, enabling enhanced interaction with the cobalt(III) center.

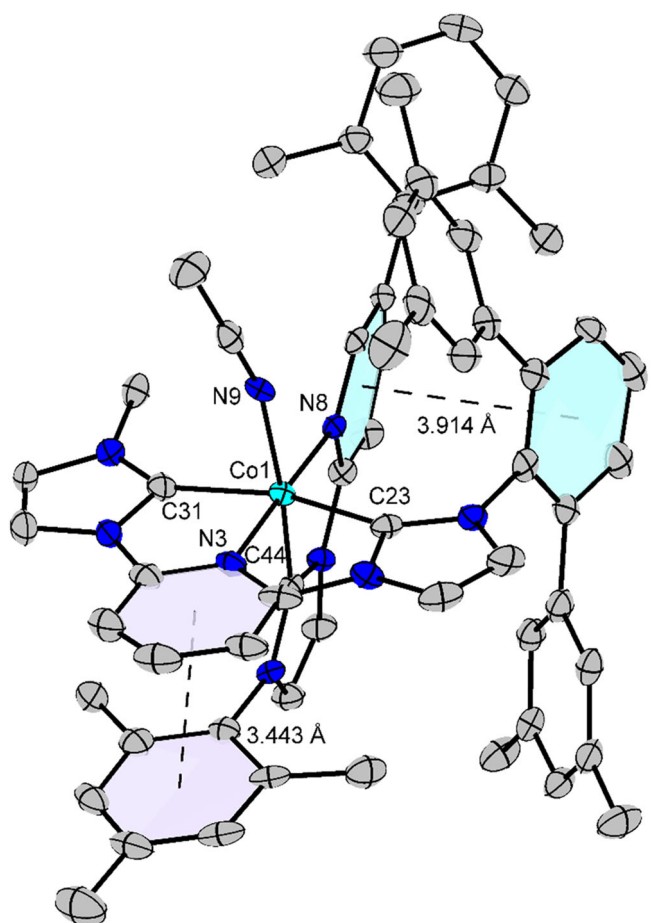

**Fig. 4 | X-ray crystal structure of Δ-CoCat4.** ORTEP drawing with 50% probability thermal ellipsoids. Solvent, three hexafluorophosphate counterions and the hydrogen atoms are omitted for clarity.

Interestingly, when the irradiation was performed in CH$_2$Cl$_2$ in the absence of additional coordinating ligands, we observed an unexpected decomposition. Specifically, exposing *rac*-**CoCat1** to blue LEDs for 20 hours at room temperature produced biphenyl **3** in 38% yield, apparently originating from the BArF counteranion (Fig. 8a). Corresponding boron degradation species could also be observed by $^{19}$F and $^{11}$B NMR (see Supplementary Information, pages 57–59). This suggests that light irradiation triggers a reaction between the cobalt cation and the BArF counteranion. Indeed, several studies have reported that B−C bond cleavage and aryl transfer from BArF anions can be readily induced by strongly Lewis acidic transition metal complexes[65–68]. Based on these observations, we propose the following activation mechanism: visible-light irradiation of the cobalt(III) complex causes dissociation of the monodentate MeCN ligand, exposing a highly Lewis acidic cobalt center. This Lewis acid center subsequently activates a nearby BArF counteranion, facilitating the transfer of a 3,5-bis(trifluoromethyl)phenyl group to the cobalt center. The resulting cobalt-aryl bond is labile and undergoes homolytic cleavage, generating a pentacoordinate cobalt(II) complex and aryl radicals, which dimerize to form biphenyl. This mechanism also explains the enhanced reactivity observed for the cobalt catalyst bearing the CF$_3$ group (compare entries 4 and 5 in Table 1), as the electron-withdrawing substituent increases the Lewis acidity of the photogenerated, coordinatively unsaturated pentacoordinate cobalt(III) intermediate. This, in turn, promotes a more efficient reaction with the BArF counteranion, leading to the formation of a pentacoordinate cobalt(II) species. Similarly, a higher concentration of BArF further facilitates this transformation (compare entries 7 and 8 in Table 1). This pentacoordinate cobalt(II) complex is proposed to act as the active catalyst for converting the isoxazole into the chiral azirine.

The following additional experiments provide strong support for the proposed mechanism. First, replacing the BArF counterions with NTF$_2$ (using *rac*-**CoCat1** with three NTF$_2$ counterions) rendered *rac*-**CoCat1** inactive for the photoinduced conversion of isoxazole **1** to azirine **2** (Fig. 8b). However, the photoactivity was fully restored upon the addition of 10 mol% NaBArF. These results clearly highlight the essential role of BArF counterions in the reaction. Second, photolyzing Δ-**CoCat3** for 1 hour at −45 °C in CH$_2$Cl$_2$, followed by the addition of the substrate and further reaction in the dark for 20 h, resulted in the formation of the azirine product in 24% yield with 91% ee (Fig. 8c). This observation supports the proposed mechanism, where light is only

## Table 1 | . Initial experiments and optimization

| entry | catalyst (mol%) | conditions[a] | T/[°C] | yield[b] | ee (%)[c] |
|---|---|---|---|---|---|
| 1 | Δ-**CoCat1** (3) | light | r.t. | 90 | 0 |
| 2 | Δ-**CoCat1** (3) | no light | r.t. | trace | - |
| 3 | Δ-**CoCat1** (3) | light | −30 | 9 | 2 (R) |
| 4 | Δ-**CoCat2** (3) | light | −30 | 23 | 42 (R) |
| 5 | Δ-**CoCat3** (3) | light | −30 | 93 | 64 (R) |
| 6 | Δ-**CoCat3** (3) | light | −45 | 82 | 82 (R) |
| 7 | Δ-**CoCat3** (3) | light | −50 | 58 | 83 (R) |
| 8 | Δ-**CoCat3** (3) | light, NaBArF[d] | −50 | 96 | 80 (R) |
| 9 | Δ-**CoCat3** (6) | light, NaBArF[d] | −60 | 78 | 91 (R) |
| 10 | Λ-**CoCat3** (3)[e] | light | −50 | 91 | 80 (S) |

[a]Standard condition: Substrate **1** (0.04 mmol) in CH$_2$Cl$_2$ (0.1 M) with cobalt catalyst (3-6 mol%) was stirred at the indicated temperature under nitrogen and irradiation with blue LEDs (24 W) for 24 h. Deviations from these standard conditions are shown. [b]Determined by $^1$H NMR using 1,3,5-trimethoxybenzene as internal standard. [c]Determined with the purified products by HPLC on chiral stationary phase. The absolute configuration is provided in brackets. [d]NaBArF, 0.004 mmol (0.1 equiv.). [e]The Λ-catalyst was synthesized with three BArF counterions.

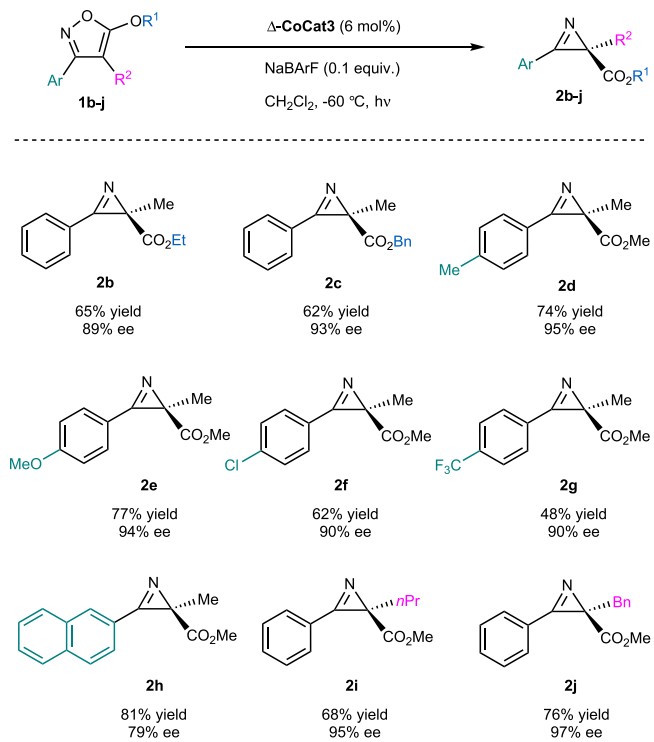

**Fig. 5 | Substrate scope for the enantioselective conversion of isoxazoles to chiral 2H-azirines.** Positions that are modified compared to the standard substrate **2** are highlighted. BArF tetrakis(3,5-bis(trifluoromethyl)phenyl)borate, ee enantiomeric excess.

required for the initial BArF-assisted generation of the catalytically active cobalt species. Finally, to confirm that the catalytically active species is indeed cobalt(II), we conducted the reaction in the dark using zinc instead of light. Treating Δ-**CoCat3** at −50 °C for 24 h with 0.6 equivalents of zinc successfully converted isoxazole **1** to azirine **2** in 82% yield and with 80% ee (Fig. 8d).

To further substantiate the proposed mechanism, we monitored the change in the absorption spectra of a 0.8 mM solution of *rac*-**CoCat3** in $CH_2Cl_2$ solution upon irradiation with blue LED ($\lambda_{max} = 467$ nm). The initial absorption spectrum (black line in Fig. 6c) evolved, and a new band peaked at 520 nm appeared (red line in Fig. 6c, after 30 min of irradiation). This change can be attributed to the photodissociation of MeCN ligand, followed by the reduction of Co(III) to Co(II) in the presence of the BArF counterion. Upon air-equilibration, the band peaked at 520 nm immediately disappeared (green line in Fig. 6d), demonstrating a fast reoxidation to Co(III); a slow recovery of the initial spectrum was observed, likely related to the regeneration of Co(III) catalyst in the dark (dashed blue line in Fig. 6d).

On the other hand, when the experiment was conducted in MeCN, no changes were observed in the absorption spectrum of *rac*-**CoCat3** (Supplementary Fig. 20), indicating that photodissociated MeCN rapidly re-coordinates due to its high concentration. Similarly, no change was observed when the experiment was performed in air-equilibrated $CH_2Cl_2$ solution because the produced Co(II) center was rapidly restored by molecular oxygen to the inert Co(III) complex. Replacing the BArF counterion with the inert $NTf_2$ counterion also had no effect on the absorption spectrum which indicates no Co(II) species were formed without BArF counterion (Supplementary Fig. 21). In the presence of substrate **1**, absorption changes similar to those reported in Fig. 8a were registered (Supplementary Fig. 22), although the steady-state concentration of produced Co(II) was lower because of the catalytic cycle that involves the substrate.

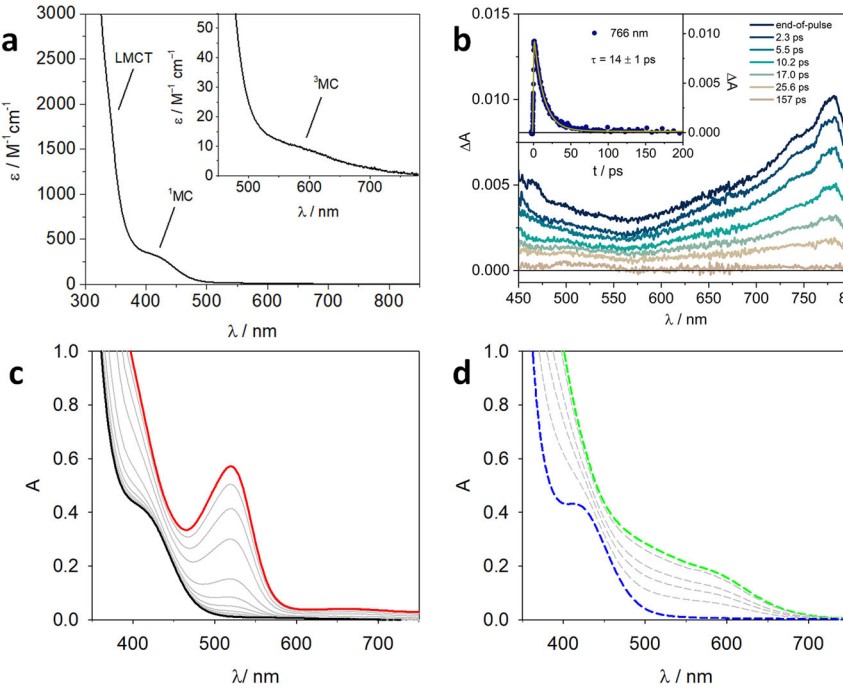

**Fig. 6 | Photophysical experiments. a** Ground-state absorption spectrum of *rac*-**CoCat3** in dichloromethane at room temperature. **b** Transient absorption spectra of *rac*-**CoCat3** in dichloromethane at the end-of-pulse and at different delays. The inset reports the ΔA decay at 766 nm and the relative fitting (line). Excitation at 350 nm ($A_{350} = 0.2$, 0.2 cm optical path, 8 μJ/pulse). **c** Absorption spectra of a

0.8 mM solution of *rac*-**CoCat3** (black line) in dichloromethane under inert atmosphere and evolution upon irradiation at 467 nm: the spectrum obtained after 30 minutes of irradiation is reported in red. **d** Time-evolution in air-equilibrated solution: the dashed green spectrum is registered immediately after air equilibration and the dashed blue line after 1 day.

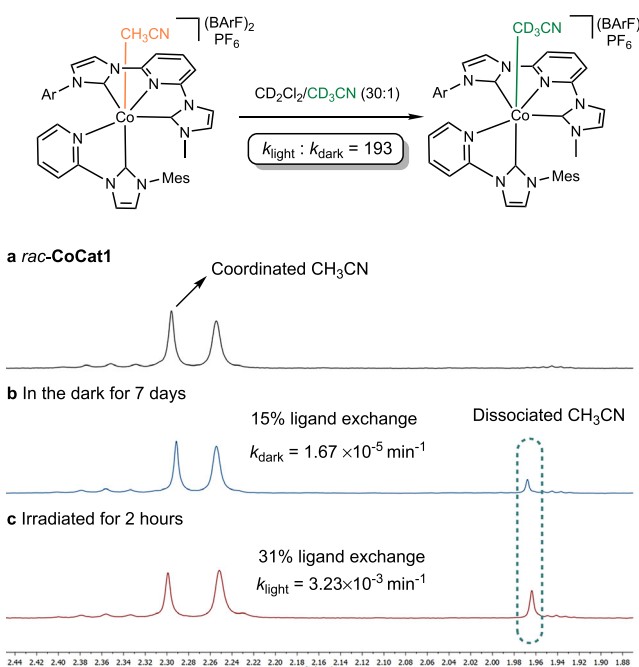

**Fig. 7 | Light-induced ligand substitution.** Excerpts of ¹H-NMR spectra of *rac*-**CoCat1** in CD₂Cl₂/CD₃CN (30:1) under different conditions: (**a**) Right after dissolution; (**b**) in the dark after 7 days; (**c**) after irradiation with blue LEDs for 2 h. BArF tetrakis(3,5-bis(trifluoromethyl)phenyl)borate.

**Fig. 8 | Mechanistic experiments. a** Identification of the biphenyl byproduct **3**. **b** BArF-dependence on the catalysis. **c** Light-preactivation of the catalyst. **d** Zinc activation of the catalyst. BArF tetrakis(3,5-bis(trifluoromethyl)phenyl)borate, NTf₂ bistriflimide.

**Fig. 9 | Proposed mechanism.** The proposed cobalt intermediates involved in the mechanism carry the following charges: intermediate **A** is tricationic, while intermediates **B** through **F** are dicationic. BArF tetrakis(3,5-bis(trifluoromethyl)phenyl) borate.

Combining all the results discussed above, the following plausible mechanism is proposed (Fig. 9). The initiation begins with the cobalt(III) precatalyst, which undergoes photoinduced dissociation of the MeCN ligand to form the pentacoordinated intermediate **A**. This tricationic species acts as a strong Lewis acid, facilitating activation of the C − B bond of a BArF counterion as part of the solvated ion pair, leading to the formation of the cobalt-aryl intermediate **B**, a transformation

with established precedent in which the oxidation state at the cobalt is unchanged. Intermediate **B** then undergoes homolytic bond cleavage, likely induced by light, to produce the pentacoordinated open-shell chiral-at-cobalt(II) complex **C**, which serves as the active catalyst for the rearrangement of the isoxazole to the chiral 2*H*-azirine. It is important to note that the homolytic cleavage of the cobalt–aryl bond represents a reduction at the cobalt center, resulting in the formation of a reactive metal center. The catalytic cycle proceeds then as follows, which is established in the literature[54,69]: the isoxazole coordinates to complex **C**, forming intermediate **D**, after which a single-electron transfer (SET) occurs from the cobalt center to the coordinated isoxazole, yielding the cobalt(III)-coordinated iminyl radical **E**. This radical subsequently undergoes a stereocontrolled cyclization to generate the cobalt(III)-coordinated azirine **F**. The asymmetric induction occurs in this step (**E → F**) and apparently requires a very bulky aryl substituent on the pincer ligand. Finally, a back-electron transfer and release of the 2*H*-azirine product regenerates the active cobalt(II) catalyst, enabling continuation of the catalytic cycle.

In summary, we here presented a previously elusive example of a (photo)reactive chiral-at-cobalt catalyst, more than a century after Alfred Werner's groundbreaking discovery of a chiral cobalt complex exhibiting exclusively metal-centered chirality. This was achieved using a (3 + 2 + 1) ligand system, which combines a meridional tridentate, a bidentate, and a monodentate ligand. Metal-centered chirality arises in this system when the meridional tridentate ligand is unsymmetric. While octahedral, hexacoordinated cobalt(III) complexes are typically inert, we discovered a convenient and mild BArF-counterion-assisted photoreduction: light triggers the conversion of

the inert cobalt(III) precatalyst into a catalytically active cobalt(II) complex. Photoinduced asymmetric cobalt catalysis in which the cobalt complex serves as both the chromophore and chiral catalyst are rare and just a recent development[70,71]. To the best of our knowledge, the BArF-assisted cobalt photoactivation is unprecedented. Remarkably, the reactive cobalt(II) intermediate retains its metal-centered chirality, allowing it to catalyze the highly enantioselective rearrangement of an isoxazole to a chiral azirine. The application of the surprising but useful BArF-assisted photoactivation is currently investigated in our laboratory for other catalytic processes.

## Methods

### Synthesis of non-racemic cobalt auxiliary complexes

A mixture of *rac*-**CoBr1-3** (0.15 mmol), (*S*)-methoxyphenylacetic acid (200 mg, 1.20 mmol) and $K_2CO_3$ (84 mg, 0.60 mmol) in $CH_3CN$/EtOH (3:1, 16 mL) was heated at 70 °C for 18 h under air. The reaction mixture was concentrated to dryness. The residue material was dissolved in $CH_2Cl_2$ and washed with $H_2O$. The organic layer was dried over MgSO4, filtered, evaporated under reduced pressure to dryness, and washed with $Et_2O$ to afford a yellow solid. The solid was dissolved in $CH_2Cl_2$ (10 mL) and then NaBArF (337 mg, 0.38 mmol) was added. The mixture was stirred at room temperature overnight under air. Then, $CH_2Cl_2$ (10 mL) was added and the solution was filtered through a plug of Celite. The filtrate was concentrated to dryness. The residue was subjected to flash silica gel chromatography ($CH_2Cl_2$/ $CH_3OH$/sat. solution of $KPF_6$ in MeCN = 200:10:1) or purified by preparative layer chromatography ($CH_2Cl_2$/$CH_3OH$ = 20:1) to obtain both diastereomers.

### Synthesis of Non-Racemic Cobalt Catalysts

To a solution of Δ-(*S*)-**CoAux1-3** (0.05 mmol) or Λ-(*S*)-**CoAux1-3** (0.05 mmol) in $CH_3CN$ (5 mL) was added $NH_4PF_6$ (0.75 mmol, 122 mg) and the tube sealed. The resulting mixture was stirred at 70 °C for 24 hours. The reaction mixture was evaporated to dryness, dissolved in $CH_2Cl_2$ (30 mL) and washed with $H_2O$ (three times 15 mL). The organic layer was dried over $MgSO_4$, filtered and evaporated under reduced pressure to dryness. The obtained solid was washed with cold $Et_2O$ and dried under vacuo to provide a yellow solid.

### General Procedure for Asymmetric Catalysis

A dried 5 mL Schlenk tube was charged with the Δ-**CoCat3** (5.1 mg, 6 mol%), substrate (0.03 mmol) and NaBArF (2.7 mg, 0.003 mmol, 0.1 equiv.). The tube was purged with nitrogen, and $CH_2Cl_2$ (0.3 mL, 0.1 M) was added via a syringe. The reaction mixture was degassed using three freeze-pump-thaw cycles. The mixture was stirred at −50 °C for 30 minutes. The vial was then positioned approximately 15 cm from a 24 W blue LEDs lamp and stirred under irradiation at −60 °C for 24 hours. Upon completion of the irradiation, the reaction mixture was diluted with cold *n*-hexane, and 1,3,5-trimethoxybenzene was added as an internal standard. The solution was passed through a short silica gel column and eluted with $CH_2Cl_2$. The combined eluents were concentrated under reduced pressure, and the crude residue was analyzed by [1]H NMR spectroscopy to determine the yield. Then, the entire mixture was collected and purified by flash chromatography on silica gel (*n*-hexane/EtOAc=5:1) to afford the product. The enantiomeric excess was determined by HPLC analysis using a chiral stationary phase.

### Photophysical measurements

Optically diluted solutions with concentrations in the order of $10^{-3}$ or $10^{-6}$ M were prepared in spectroscopic or HPLC grade solvents for steady-state and time-resolved absorption analysis. Absorption spectra were recorded at room temperature on a Varian Cary 300 spectrophotometer with 1 cm or 0.2 cm quartz cuvettes. Degassed solutions were prepared via 4 consecutive freeze-pump-thaw cycles and spectra were taken using a home-made Schlenk quartz cuvette; alternatively, $N_2$-saturated solutions were prepared in a glove box using the same glassware. The estimated experimental errors are 2 nm on the band maximum and 5% on the molar absorption coefficient. Irradiation of samples in $CH_3CN$ or $CH_2Cl_2$ were performed at room temperature on thoroughly stirred $N_2$-saturated solutions by using a Kessil lamp at 467 nm (40 W) or blue LED (LED Engin LuxiGenTM LZ1-10DB00, λ = 450– 480 nm) operating at 10 V and 500 mA. The sample was placed at 15 cm from the source.

### Transient absorption spectroscopy

Pump-probe transient absorption measurements were performed with an Ultrafast Systems HELIOS (HE-VIS-NIR) femtosecond transient absorption spectrometer by using, as excitation source, a Newport Spectra Physics Solstice-F-1K-230 V laser system, combined with a TOPAS Prime (TPR-TOPAS-F) optical parametric amplifier (pulse width: 100 fs, 1 kHz repetition rate) tuned at 350 nm. The pump energy on the sample was 8 μJ/pulse. The probe continuum generation was in the visible range (450–800 nm). The overall time resolution of the system is 300 fs. Air-equilibrated solutions of the sample in 0.2 cm optical path cells were analyzed under continuous stirring. Surface Xplorer software from Ultrafast Systems was used for data acquisition and analysis. The raw 3D surfaces were corrected for the chirp of the probe pulse prior to analysis. The error on the lifetime derives from the fitting procedure.

## Data availability

The experimental procedures, characterization data, NMR spectroscopic data, and photophysical experiments generated in this study are provided in the Supplementary Information. All data are also available from the corresponding authors upon request. The X-ray crystallographic coordinates for structures reported in this study have been deposited at the Cambridge Crystallographic Data Centre (CCDC), under deposition numbers 2452612 (**Co-bpq**) and 2425622 (Δ-**CoCat4**). These data can be obtained free of charge from The Cambridge Crystallographic Data Centre via www.ccdc.cam.ac.uk/ data_request/cif.

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

## Acknowledgements
This project has received funding from the European Research Council (ERC) under the European Union's Horizon 2020 research and innovation programme (grant agreement No. 883212 to E.M.). P.C., M.V., and A.F. acknowledge the project SUN-SPOT funded by the MIUR Progetti di Ricerca di Rilevante Interesse Nazionale (PRIN) Bando 2022 (grant No. 2022JA3PSC). S.-Y.Y. thanks the China Scholarship Council, Guangdong Basic and Applied Basic Research Foundation (No. 2023A1515110790) for financial support. B.V. acknowledges the Italian CNR (project Light-Induced Processes "LIP"). We thank Dr. Xiulan Xie (Fachbereich Chemie, Philipps-Universität Marburg) for assistance with NMR measurements.

## Author contributions
E.M., S.-Y.Y. and P.C. wrote the manuscript. E.M. and S.-Y.Y. conceived the project and devised the synthetic and catalytic experiments. M.V., A.F., B.V. and P.C. conceived, performed and analyzed the photo-physical experiments. Y.Z. performed catalytic experiments. S.I.I. per-formed the X-ray crystallographic analysis.

## Funding

## Competing interests
The authors declare no competing interests.
