## [Transparent Peer Review file · Nature Communications]

Cobalt catalyst with exclusive metal-centered chirality for asymmetric photocatalysis

Corresponding Author: Professor Eric Meggers

Version 0:

Reviewer comments:

Reviewer #1

(Remarks to the Author)

In this manuscript entitled "Cobalt catalyst with exclusive metal-centered chirality for asymmetric photocatalysis", the authors designed new chiral-at-cobalt(III) (pre)catalysts (3+2+1 design) that have only achiral ligands. The cobalt centers act as not only stereogenic centers as well as active catalytic centers for asymmetric photo-catalysis. They investigated mechanistic study carefully, and clarified BArF-assisted photoreduction to generate active cobalt(II) complexes. This work would give a new insight to design chiral-at-metal complexes.

Though the data in the manuscript are well summarized about the mechanistic studies (purity and stability of the complexes, and reaction mechanism), one significant problem is limitation of substances. Only one substance, 5-methoxy-4-methyl-3-phenylisoxazole was investigated. Readers would be interested in substrate scope of substrates from the view points of selectivity and reactivity.

In my opinion, it requires major revision before ready for publication.

Additional Comment:

In the Supplementary Information page 36, a small peak can be seen around 7.05 ppm. I want to confirm i) whether this small peak comes from Δ -CoCat3 or not, ii) ratio of Λ -CoCat3 and this small peak.

Reviewer #2

(Remarks to the Author)

This article by Ceroni and Meggers reports the conception, synthesis and catalytic evaluation of a chiral-at-metal cobalt complex with a new ligand design. By using a combination of one tridentate, one bidentate and one monodentate ligand (referred to as 3+2+1 strategy), they prepare a chiral cobalt Co(+III) complex without any chiral ligand. Both enantiomers are accessed through separation of the diastereomers arising from monodentate ligand exchange between acetonitrile and (S)-methoxyphenylacetic acid. Through this pathway, the authors generate a family of octahedral cobalt complexes that turn out to be efficient photocatalysts and perform the conversion of isoxazole into a chiral azirine with good yields and ees up to 91%. Mechanistic investigations based on photophysical properties determination, transient absorption spectroscopy, ¹H NMR studies and UV-vis spectroscopy shed light on the mechanism of this transformation.

This is an exciting breakthrough in the field of enantioselective photocatalysis and will undoubtedly have a strong impact on its future directions. The logic and writing mechanics are clear, detailed and concise, and enable easy reading. However, some parts of the study need further clarification, notably the mechanistic section. This article should be suitable for publication in Nature Communications provided the following points are addressed:

- In figure 1, the presence of both oxidation states symbolized as charges +3 instead of +III on the metal center and either/or

as a general charge is confusing. The top complex has no overall charge but the Co is a Co^{3+} cation while the complex below has a Co^{2+} cation and an overall $2+$ charge ascribed, for the sake of homogeneity a single notation would be best. Also, the molarity of the BArF counterion is not mentioned in the top part of the figure.

- In the optimization section, the yield improvement between entries 4 and 5 is quite dramatic (23% to 93%). How do the authors interpret such strong effect of the CF₃ group? Similar question between entries 7 and 8, why does raising the loading of "free" BArF induce such a strong effect?

Based on combined insights, the authors suggest a mechanistic pathway (Fig. 9) that raises some questions/comments:

- Could the authors specify the charges of the complexes and intermediates mentioned? Also, since roman numbers are already being used for metallic oxidation states, referring to the intermediates with letters might be helpful for the discussion.

- An initial photoinduced ligand decooordination step followed by an aryl transfer from the BArF anion and subsequent homolytic cleavage of the arylmetal bond leads to a $\text{Co}(\text{+II})$ center (intermediate III) in three steps. While the decooordination event and first step is well evidenced and observed by the authors, the formation of intermediate II is not so clear. What is the nature of the electron transfer or the redox reaction in this step?

- Departure of MeCN is a L ligand dissociation and therefore does not change the oxidation state nor charge, this leads to intermediate I represented as $\text{Co}(\text{+III})$. However, coordination of an X-type aryl ligand should change the oxidation state of the metal and intermediate II should then not be represented as $\text{Co}(\text{+III})$ (provided all redox events are metal-based). Overall the reaction with BArF seems slightly unclear, have the authors considered alternative pathways for the formation of the bis-aryl(CF₃)₂ product and what happens to the remaining boron fragment? Do they observe other byproducts?

- On intermediate III an unpaired electron is represented. $\text{Co}(\text{+II})$ is a d^7 open-shell state and thus inherently a radical, the following addition of 1 (represented as a L ligand) leads to IV so where did the radical go?

- Arrows should be made distinct between simple arrows and fish-hooks for single electron transfer, should rearrangements in IV and VI be fish-hooks?

- Considering the mechanism proposed by the authors, it would be interesting to investigate the localization of the spin density in the radical $\text{Co}(\text{+II})$ (species III in fig. 9) as this intermediate probably has different spin distributions depending on the ligand, which is probably mirrored by the yield difference observed in the optimization table between entries 4 and 5. The introduction of CF₃ group probably as a strong impact on the energy levels of the ligand-based MOs.

- How is the control exerted in the step responsible for enantioselectivity? Has the Lambda isomer been tested and does it give the other enantiomer and/or the same level of enantioselectivity?

To summarize, this work is of very high interest and with potential strong implications but these should be expanded upon to clarify the reaction mechanism. Insights could guide future developments of this exciting strategy.

Reviewer #3

(Remarks to the Author)

In this manuscript, Ceroni, Meggers, and collaborators present the first example of a chiral-only-at-cobalt asymmetric catalyst activated through a rather unexpected mechanism. They describe a surprising photoinduced activation (reduction) of an inert chiral-at-metal $\text{Co}(\text{III})$ compound to form a reactive $\text{Co}(\text{II})$ complex. This activation is triggered by the presence of a non-innocent BArF counterion, which unexpectedly serves as a sacrificial reducing agent. This work exemplifies how the most fascinating discoveries often originate from serendipitous reactivities, provided a wise observer supervises them. The work is scholarly presented. The authors offer many experimental evidences to support the operating mechanisms proposed, which are discussed in the main text and complemented by comprehensive supplementary material. Even if the results are not outstanding in terms of catalytic activity (when compared with other Fe or Ru-based catalysts from the same authors), the originality and broad applicability of the work, together with the "poetical" fact that it is the first example of a Werner-type cobalt complex acting as an asymmetric catalyst, makes it deserve publication in Nature Communications. This work will probably open a new venue for other asymmetric photochemical reactions.

Some minor suggestions, comments, and typo corrections are listed here. These comments are intended to help the authors further improve the quality of the work described.

1) As the authors mention, the complexes described in this work are constructed using achiral ligands within a (3+2+1) coordination geometry. In all cases, the NHC fragment of the bidentate ligand (2) occupies a position trans to the "reactive site" (1). This selectivity is relevant, as it reduces the number of possible stereoisomers, and it may deserve comment in the main text. Related to this, considering the σ -donating character of the NHC fragment of the bidentate ligand (trans to the "reactive site"), to which extent is this fragment "responsible" for the observed reactivity? For instance, does one expect a similar reactivity if a 2,2'-bipyridine was used as bidentate?

2) The authors state that the diastereoisomers CoAux1-4 are separated by preparative TLC (PLC) and/or column chromatography. It would be worth specifying the R_f of each diastereomer? Since these compounds are described as BArF salts, it seems unclear why a saturated solution of KPF₆ was used as a mobile phase additive. Could the authors comment on this?

3) Since both enantiomers of Cat1-3 have been isolated, did the authors test the catalytic performance of both enantiomers, for instance, under optimized conditions? Even if it is trivial, as the enantiopurity has been checked by ECD, it is always nice to confirm that they produce enantiomeric products (and permits evaluating the magnitude of the experimental error).

4) Considering that the authors describe the X-ray structure of the -Co-Cat-4 (containing a bulky 2,6-Me₂Ph group as R substituent on the pyridyl-NHC ligand, it is rather surprising that it has not been tested under catalytic conditions. One can notice that it has been synthesized as tris-hexafluorophosphate salt, but still, it seems worth reporting its catalytic activity with exogenous BARF added or isolating the corresponding bis-BARF/PF₆ salt, to evaluate the effect of this bulky substituent on the bidentate ligand.

5) One may guess that (in addition to the fluorinated bisphenol) the other byproduct of the photoactivation is the fluorinated borane B(ArF)₃. Did the authors detect its presence in the reaction mixture (by ¹¹B{¹H}-NMR, for instance)? Related to this, the authors missed a very recent reference on metal-mediated B-C bond cleavage of BARF that should be included (Organometallics 2025, 44, 1, 340–346). In this publication, one can find a description of the ¹¹B{¹H}-NMR and ¹⁹F{¹H}-NMR of the formed borane mentioned above.

Typo mistakes:

First paragraph of the methods section. It reads “The solid was dissolved in 10 mL CH₂Cl₂ (10 mL)...” (10 mL) should be removed.

When describing the catalytic results (page 8) it reads “Interestingly, the addition of NaBARF (0.1 equiv) increased the yield to 80% (entry 8)”. In entry 8 the yield is 96%.

Version 1:

Reviewer comments:

Reviewer #1

(Remarks to the Author)

Thank your kind response to my review.

Scope of substrate:

Thank you for your trial about substrate scope.

Not only the improvement of enantioselectivity (up to 97%),

Wide substrate scope was realized in this system.

Though the bulky substituent (2-naphthyl group) gave slightly lower selectivity.

Purity of CoCat3

Thank for checking the enantiopurity of CoCat3.

The peak of ΔCoCat3 is small but I think it is not negligible.

If you check (non-)linear effect of the catalysts, the result may support your proposed mechanism with a monomeric catalyst. (additional comment).

The manuscript has been revised well, so I think this manuscript is acceptable to Nature Communications.

Reviewer #2

(Remarks to the Author)

Having read the revised manuscript for this article, I am satisfied by the changes made to the manuscript as well as the detailed responses that the authors provide for the several points raised.

I support publication of this revised version in Nature Communications.

Reviewer #3

(Remarks to the Author)

The authors properly addressed all the concerns I raised in my previous revision, and I do not have further comments.

Therefore, I support the publication of this work in its current form.

Point-By-Point Response to the Comments of the Reviewers

Reviewer #1 (Remarks to the Author):

In this manuscript entitled "Cobalt catalyst with exclusive metal-centered chirality for asymmetric photocatalysis", the authors designed new chiral-at-cobalt(III) (pre)catalysts (3+2+1 design) that have only achiral ligands. The cobalt centers act as not only stereogenic centers as well as active catalytic centers for asymmetric photo-catalysis. They investigated mechanistic study carefully, and clarified BArF-assisted photoreduction to generate active cobalt(II) complexes. This work would give a new insight to design chiral-at-metal complexes.

Our response: We appreciate these positive comments from reviewer #1.

Though the data in the manuscript are well summarized about the mechanistic studies (purity and stability of the complexes, and reaction mechanism), one significant problem is limitation of substances. Only one substance, 5-methoxy-4-methyl-3-phenylisoxazole was investigated. Readers would be interested in substrate scope of substrates from the view points of selectivity and reactivity.

Our response: As a response to this request we have performed a substrate scope and added it to the revised manuscript as the new Fig. 5. Experimental procedures and chiral HPLC traces are added to the revised Supporting Information (new Section 7.2, pages 48-53 and Section 11, pages 70-80). The substrate scope demonstrates that this catalytic asymmetric rearrangement is applicable to substrates with modified electronic and steric properties. Interestingly, ee values of up to 97% ee were obtained!

In my opinion, it requires major revision before ready for publication.

Additional Comment:

In the Supplementary Information page 36, a small peak can be seen around 7.05 ppm. I want to confirm i) whether this small peak comes from Δ -CoCat3 or not, ii) ratio of Δ -CoCat3 and this small peak.

Our response: We apologize for not making this more clear. Yes, the small peak results from small amounts of Δ -CoCat3. In the revised Supplementary Information (Fig. S4, page 37) we now label this minor peak and explain how we derived at the enantiomeric excess (SI, page 36).

Reviewer #2 (Remarks to the Author):

This article by Ceroni and Meggers reports the conception, synthesis and catalytic evaluation of a chiral-at-metal cobalt complex with a new ligand design. By using a combination of one tridentate, one bidentate and one monodentate ligand (referred to as 3+2+1 strategy), they prepare a chiral cobalt Co(+III) complex without any chiral ligand. Both enantiomers are accessed through separation of the diastereomers arising from monodentate ligand exchange between acetonitrile and (S)-methoxyphenylacetic acid. Through this pathway, the authors generate a family of octahedral cobalt complexes that turn out to be efficient photocatalysts and perform the conversion of isoxazole into a chiral azirine with good yields and ees up to 91%. Mechanistic investigations based on photophysical properties determination, transient absorption spectroscopy, ^1H NMR studies and UV-vis spectroscopy shed light on the mechanism of this transformation.

This is an exciting breakthrough in the field of enantioselective photocatalysis and will undoubtedly have a strong impact on its future directions. The logic and writing mechanics are clear, detailed and concise, and enable easy reading. However, some parts of the study need further clarification, notably the mechanistic section. This article should be suitable for publication in Nature Communications provided the following points are addressed:

Our response: We appreciate these positive comments from reviewer #2 and addressed the mechanistic points raised below.

- In figure 1, the presence of both oxidation states symbolized as charges +3 instead of +III on the metal center and either/or as a general charge is confusing. The top complex has no overall charge but the Co is a Co^{3+} cation while the complex below has a Co^{2+} cation and an overall 2+ charge ascribed, for the sake of homogeneity a single notation would be best. Also, the molarity of the BARF counterion is not mentioned in the top part of the figure.

Our response: We apologize for the confusion and concur with reviewer #2 that a more consistent notation is preferable. To address this, the revised Figure 1 now displays both the oxidation state of cobalt and the overall charge of the complexes. Additionally, we have adjusted the molarity of the counterions for clarity and accuracy.

- In the optimization section, the yield improvement between entries 4 and 5 is quite dramatic (23% to 93%). How do the authors interpret such strong effect of the CF_3 group? Similar question between entries 7 and 8, why does raising the loading of "free" BARF induce such a strong effect?

Our response: We thank reviewer #2 for these insightful questions. In our opinion, the effects are fully consistent with the proposed mechanism. To make this clear, we added the following statement to page 13 of the revised manuscript: "This mechanism also explains the enhanced reactivity observed for the cobalt catalyst bearing the CF_3 group (compare entries 4 and 5 in Table 1), as the electron-withdrawing substituent increases the Lewis acidity of the photogenerated, coordinatively unsaturated pentacoordinate cobalt(III) intermediate. This, in turn, promotes a more efficient reaction with the BARF counteranion, leading to the formation of a pentacoordinate cobalt(II) species. Similarly, a higher concentration of BARF further facilitates this transformation (compare entries 7 and 8 in Table 1)."

Based on combined insights, the authors suggest a mechanistic pathway (Fig. 9) that raises some questions/comments:

- Could the authors specify the charges of the complexes and intermediates mentioned? Also, since roman numbers are already being used for metallic oxidation states, referring to the intermediates with letters might be helpful for the discussion.

Our response: We followed these suggestions and have adjusted Fig. 10 (previously Fig. 9) as follows.

a.) Intermediates are now labeled with capital letters (A-F instead of previously I-VI).

b.) We prefer not to show the overall charge of the individual intermediates as this might be confusing. Instead, we added a sentence to the legend in which we state the individual charges: “The proposed cobalt intermediates involved in the mechanism carry the following charges: intermediate A is tricationic, while intermediates B through F are dicationic.”

- An initial photoinduced ligand decooordination step followed by an aryl transfer from the BARF anion and subsequent homolytic cleavage of the arylmetal bond leads to a Co(+II) center (intermediate III) in three steps. While the decooordination event and first step is well evidenced and observed by the authors, the formation of intermediate II is not so clear. What is the nature of the electron transfer or the redox reaction in this step?

Our response: We thank reviewer #2 for this insightful question. Despite considerable efforts, we were unable to isolate the cobalt–aryl intermediate II (reabeled as intermediate B in the revised manuscript), likely due to its insufficient stability. However, its formation via reaction with BARF, as well as its subsequent decomposition to a cobalt(II) species, are well-established in the literature. The decomposition step (B → C) does not involve electron transfer, but rather proceeds via homolytic cleavage of the Co–C(aryl) bond, which formally results in a change in the oxidation state at the cobalt center. To make this more clear we added the following statement to the summary of the mechanism on page 16: “It is important to note that the homolytic cleavage of the cobalt–aryl bond represents a reduction at the cobalt center, resulting in the formation of a reactive metal center.”

- Departure of MeCN is a L ligand dissociation and therefore does not change the oxidation state nor charge, this leads to intermediate I represented as Co(+III). However, coordination of an X-type aryl ligand should change the oxidation state of the metal and intermediate II should then not be represented as Co(+III) (provided all redox events are metal-based). Overall the reaction with BARF seems slightly unclear, have the authors considered alternative pathways for the formation of the bis-aryl(CF₃)₂ product and what happens to the remaining boron fragment? Do they observe other byproducts?

Our response:

Regarding the oxidation state of intermediate II (reabeled as intermediate B): We respectfully disagree with the statement made by reviewer #2. The oxidation state of intermediate B is correctly assigned as III. The aryl group is formally transferred to the cobalt center as an aryl anion, thereby leaving the oxidation state of cobalt unchanged. To clarify this point, we have added the following remark to the summary of the mechanism on pages 15/16: “...leading to the formation of the cobalt–aryl intermediate **B**, a transformation with established precedent in which the oxidation state of cobalt remains unchanged.”

Regarding an alternative mechanism: The proposed BARF mechanism has literature precedent and is consistent with all experimental results. The extremely short lifetime of the photoexcited cobalt(III) complex (as mentioned on page 10) renders an intermolecular electron transfer reaction very unlikely.

What happens to the remaining boron fragment? As a response to this question we performed additional photoinduced catalytic experiments and analyzed the crude reaction mixture by ^{11}B and ^{19}F NMR, following literature precedent (*Organometallics* **2024**, *44*, 340 and *Organometallics* **2022**, *41*, 1475). As a result, in addition to the BArF anion, two boron species were detected which could be assigned to the boronic acid $\text{B}(\text{ArF})_2\text{OH}$ and its hydroxide adduct $[\text{B}(\text{ArF})_2(\text{OH})_2]^-$. Such oxidative or hydrolytic degradation of unstable $\text{B}(\text{ArF})_3$ has been reported (*Organometallics* **2024**, *44*, 340 and *Organometallics* **2022**, *41*, 1475) and is consistent with our mechanism. We added these results to the revised SI (Section 8.4, pages 58-60) and added the following statement to the revised manuscript on page 12: “Corresponding boron degradation species could also be observed by ^{19}F and ^{11}B NMR (see Supporting Information for more details).”

- On intermediate III an unpaired electron is represented. Co(+II) is a d7 open-shell state and thus inherently a radical, the following addition of 1 (represented as a L ligand) leads to IV so where did the radical go?

Our response: We thank reviewer #2 for identifying this oversight. In the revised mechanism, we now indicate that intermediate IV (reabeled as intermediate D) possesses radical character.

- Arrows should be made distinct between simple arrows and fish-hooks for single electron transfer, should rearrangements in IV and VI be fish-hooks?

Our response: We followed the standard convention, in which radical reactions are depicted using fishhook arrows, such as the reaction of intermediate V (reabeled as E), while electron transfer steps are represented with standard full arrows, but highlighted in red for clarity (intermediates IV and VI, reabeled as D and F).

- Considering the mechanism proposed by the authors, it would be interesting to investigate the localization of the spin density in the radical Co(+II) (species III in fig. 9) as this intermediate probably has different spin distributions depending on the ligand, which is probably mirrored by the yield difference observed in the optimization table between entries 4 and 5. The introduction of CF_3 group probably has a strong impact on the energy levels of the ligand-based MOs.

Our response: We thank the reviewer for this insightful comment. However, we believe that a detailed DFT study would be more appropriate for a separate investigation. As for the CF_3 effect, we consider it to be fully consistent with the proposed mechanism, as it enhances the Lewis acidity of the cobalt center, thereby accelerating the reaction with the BArF counteranion, as discussed above.

- How is the control exerted in the step responsible for enantioselectivity? Has the Lambda isomer been tested and does it give the other enantiomer and/or the same level of enantioselectivity?

Our response:

Regarding the stereocontrol responsible for the enantioselectivity: We added the following statement to the summary of the mechanism on page 16: “The asymmetric induction occurs in this step (E → F) and apparently requires a very bulky aryl substituent on the pincer ligand.”

Regarding the other enantiomer: We added a new entry to Table 1 (entry 10) using the mirror-image catalyst Λ -CoCat3 and we added the following statement to the text on page 8: “Finally, employing the mirror-image catalyst Λ -CoCat3 (entry 10) provided the enantiomeric azirine product with the same level of enantioselectivity (see comparison of entries 8 and 10).” The experimental data have been added to the revised SI (Section 2.10, page 31 and Section 7.12, pages 46-47).

To summarize, this work is of very high interest and with potential strong implications but these should be expanded upon to clarify the reaction mechanism. Insights could guide future developments of this exciting strategy.

Our response: We thank this reviewer for the insightful comments and questions, which have significantly contributed to the improvement of the revised mechanistic section.

Reviewer #3 (Remarks to the Author):

In this manuscript, Ceroni, Meggers, and collaborators present the first example of a chiral-only-at-cobalt asymmetric catalyst activated through a rather unexpected mechanism. They describe a surprising photoinduced activation (reduction) of an inert chiral-at-metal Co(III) compound to form a reactive Co(II) complex. This activation is triggered by the presence of a non-innocent BArF counterion, which unexpectedly serves as a sacrificial reducing agent. This work exemplifies how the most fascinating discoveries often originate from serendipitous reactivities, provided a wise observer supervises them.

The work is scholarly presented. The authors offer many experimental evidences to support the operating mechanisms proposed, which are discussed in the main text and complemented by comprehensive supplementary material.

Even if the results are not outstanding in terms of catalytic activity (when compared with other Fe or Ru-based catalysts from the same authors), the originality and broad applicability of the work, together with the "poetical" fact that it is the first example of a Werner-type cobalt complex acting as an asymmetric catalyst, makes it deserve publication in Nature Communications. This work will probably open a new venue for other asymmetric photochemical reactions.

Our response: We are greatly appreciate the very positive feedback from reviewer #3.

Some minor suggestions, comments, and typo corrections are listed here. These comments are intended to help the authors further improve the quality of the work described.

1) As the authors mention, the complexes described in this work are constructed using achiral ligands within a (3+2+1) coordination geometry. In all cases, the NHC fragment of the bidentate ligand (2) occupies a position trans to the "reactive site" (1). This selectivity is relevant, as it reduces the number of possible stereoisomers, and it may deserve comment in the main text. Related to this, considering the σ -donating character of the NHC fragment of the bidentate ligand (trans to the "reactive site"), to which extent is this fragment "responsible" for the observed reactivity? For instance, does one expect a similar reactivity if a 2,2'-bipyridine was used as bidentate?

Our response: We thank reviewer #3 for these insightful comments and thoughtful questions.

Regarding the NHC fragment of the bidentate ligand: Reviewer #3 is correct in noting that the diastereoselectivity of the coordination of the bidentate pyridyl NHC ligand is crucial. Steric factors dictate that the NHC must coordinate trans to the "reactive site", allowing the mesityl group to stack favorably with the π -system of the tridentate ligand. In the alternative theoretical possible diastereomer, the mesityl group would clash with the substituents of the tridentate ligand, rendering this arrangement much less favorable. We added a brief discussion of this aspect to the revised manuscript on page 7 and now write: "The crystal structure also highlights an important design rationale for selecting a pyridyl NHC ligand as the bidentate fragment: steric factors require that the NHC coordinates trans to the monodentate ligand (the reactive site), enabling the mesityl group to stack favorably with the π -system of the tridentate ligand and thus governing the diastereoselectivity. In the alternative theoretical diastereomer, the mesityl group would clash with the substituents of the tridentate ligand, making this arrangement significantly less favorable."

Regarding 2,2'-bipyridine as an alternative bidentate ligand: Despite the advantages outlined above for using a pyridyl NHC bidentate ligand, we attempted to coordinate 2,2'-bipyridine or 1,10-phenanthroline for comparison purposes, but were unable to isolate any clean products. However,

coordination of 2,2'-biquinoline was successful. Interestingly, this led to the formation of a pentacoordinate cobalt(III) complex (see structure below). The resulting trigonal bipyramidal complex is achiral and therefore not useful for our purposes. Since these results further underscore the critical role of the NHC moiety in the bidentate ligand design we added the crystal structure of this complex to the Supplementary Information (Section 10.2, pages 67-69) and added the following statement to the revised manuscript on page 7: "In this context, it is noteworthy that attempts to coordinate 2,2'-bipyridine or 1,10-phenanthroline as bidentate ligands in place of the pyridyl-NHC were unsuccessful. In contrast, coordination of 2,2'-biquinoline yielded an achiral pentacoordinate complex (see Supporting Information), further underscoring the significance of the pyridyl-NHC moiety."

2) The authors state that the diastereoisomers CoAux1-4 are separated by preparative TLC (PLC) and/or column chromatography. It would be worth specifying the R_f of each diastereomer? Since these compounds are described as BArF salts, it seems unclear why a saturated solution of KPF₆ was used as a mobile phase additive. Could the authors comment on this?

Our response:

Regarding the R_f values of CoAux1-4: We added the R_f values of CoAux1-4 to the revised Supplementary Information (Section 2.5, pages 18-22).

Regarding using a saturated solution of KPF₆ for the mobile phase: Regarding the mobile phase composition, we noticed that KPF₆ does not induce anion exchange with BArF in these complexes, as the large cation preferentially retains the bulky BArF anion. The addition of KPF₆ to the acetonitrile solvent serves to increase the polarity of the eluent. This strategy is particularly valuable for separating highly polar, charged organometallic complexes, where conventional methods often fall short. We added a comment of clarification to the revised Supplementary Information on page 17.

3) Since both enantiomers of Cat1-3 have been isolated, did the authors test the catalytic performance of both enantiomers, for instance, under optimized conditions? Even if it is trivial, as the enantiopurity has been checked by ECD, it is always nice to confirm that they produce enantiomeric products (and permits evaluating the magnitude of the experimental error).

Our response: We agree that it is valuable to confirm that the mirror-image catalyst provides the mirror-image catalytic product, as to be expected. We therefore added a new entry to Table 1 (entry 10) using the mirror-image catalyst Λ -CoCat3 and we added the following statement to the text on page 8: "Finally, employing the mirror-image catalyst Λ -CoCat3 (entry 10) provided the

enantiomeric azirine product with the same level of enantioselectivity (see comparison of entries 8 and 10).”

4) Considering that the authors describe the X-ray structure of the Δ -Co-Cat-4 (containing a bulky 2,6-Me₂Ph group as R substituent on the pyridyl-NHC ligand, it is rather surprising that it has not been tested under catalytic conditions. One can notice that it has been synthesized as tris-hexafluorophosphate salt, but still, it seems worth reporting its catalytic activity with exogenous BArF added or isolating the corresponding bis-BArF/PF₆ salt, to evaluate the effect of this bulky substituent on the bidentate ligand.

Our response: CoCat4 is catalytically not active (in the presence of BArF or with BArF counterions), probably due to the high steric hindrance. However, it is the only complex that, in our hands, provided suitable crystals for X-ray diffraction. We added this information to the revised manuscript and revised Supplementary Information (Section 10.1, page 64). In the revised manuscript, we now write on page 6: “Although CoCat4 (with or without BArF counterions) proved catalytically inactive, presumably due to high steric hindrance, it was the only complex in our hands that yielded suitable crystals for single-crystal X-ray diffraction analysis.”

5) One may guess that (in addition to the fluorinated bisphenol) the other byproduct of the photoactivation is the fluorinated borane B(ArF)₃. Did the authors detect its presence in the reaction mixture (by ¹¹B{1H}-NMR, for instance)? Related to this, the authors missed a very recent reference on metal-mediated B-C bond cleavage of BArF that should be included (*Organometallics* 2025, 44, 1, 340-346). In this publication, one can find a description of the ¹¹B{1H}-NMR and ¹⁹F{1H}-NMR of the formed borane mentioned above.

Our response: We thank reviewer #3 for bringing this reference to our attention. We have added it to the revised manuscript as the new reference 68. Regarding detecting the borane side product: We performed additional photoinduced catalytic experiments and analyzed the crude reaction mixture by ¹¹B and ¹⁹F NMR, following literature precedent (*Organometallics* 2024, 44, 340 and *Organometallics* 2022, 41, 1475). As a result, in addition to the BArF anion, two boron species were detected which could be assigned to the boronic acid B(ArF)₂OH and its hydroxide adduct [B(ArF)₂(OH)₂]⁻. Such oxidative or hydrolytic degradation of unstable B(ArF)₃ has been reported (*Organometallics* 2024, 44, 340 and *Organometallics* 2022, 41, 1475) and is consistent with our mechanism. We added these results to the revised SI (Section 8.4, pages 58-60) and added the following statement to the revised manuscript on page 12: “Corresponding boron degradation species could also be observed by ¹⁹F and ¹¹B NMR (see Supporting Information for more details).”

Typo mistakes:

First paragraph of the methods section. It reads “The solid was dissolved in 10 mL CH₂Cl₂ (10 mL)...” (10 mL) should be removed. When describing the catalytic results (page 8) it reads “Interestingly, the addition of NaBArF (0.1 equiv) increased the yield to 80% (entry 8)”. In entry 8 the yield is 96%.

Our response: We thank reviewer #3 for catching this mistake which we corrected (96% yield is correct). We carefully proofread the revised manuscript and revised Supplementary Information, correcting the indicated errors as well as some additional mistakes.

Point-By-Point Response to the Comments of the Reviewers

Reviewer #1 (Remarks to the Author):

Thank your kind response to my review.

Scope of substrate:

Thank you for your trial about substrate scope.

Not only the improvement of enantioselectivity (up to 97%),

Wide substrate scope was realized in this system.

Though the bulky substituent (2-naphthyl group) gave slightly lower selectivity.

Our response: We thank reviewer #1 for this comment. Yes, the substrate scope strengthens the manuscript.

Purity of CoCat3

Thank for checking the enantiopurity of CoCat3.

The peak of Δ CoCat3 is small but I think it is not negligible.

If you check (non-)linear effect of the catalysts, the result may support your proposed mechanism with a monomeric catalyst. (additional comment).

Our response: We thank reviewer #1 for this insightful comment. Non-linear effects are often attributed to catalyst aggregation. However, the cationic nature of our catalyst makes such aggregation unlikely due to electrostatic repulsion.

The manuscript has been revised well, so I think this manuscript is acceptable to Nature Communications.

Our response: Reviewer #1 is satisfied with our revisions and supports the publication of this revised manuscript.

Reviewer #2 (Remarks to the Author):

Having read the revised manuscript for this article, I am satisfied by the changes made to the manuscript as well as the detailed responses that the authors provide for the several points raised.

I support publication of this revised version in Nature Communications.

Our response: Reviewer #2 is satisfied with our revisions and supports the publication of this revised manuscript.

Reviewer #3 (Remarks to the Author):

The authors properly addressed all the concerns I raised in my previous revision, and I do not have further comments. Therefore, I support the publication of this work in its current form.

Our response: Reviewer #3 is satisfied with our revisions and supports the publication of this revised manuscript.